# CTQWFORMER: A CTQW-BASED TRANSFORMER FOR GRAPH CLASSIFICATION

## ABSTRACT

Graph Neural Networks (GNN) and Transformer-based architectures have achieved remarkable progress in graph learning, yet they still struggle to capture both global structural dependencies and model the dynamic information propagation. In this paper, we propose CTQWformer, a hybrid graph learning framework that integrates continuous-time quantum walks (CTQW) with GNN. CTQWformer employs a trainable Hamiltonian that fuses graph topology and node features, enabling physically grounded modeling of quantum walk dynamics that captures rich and intricate graph structure information.The extracted CTQW-based representations are incorporated into two complementary modules:(i) a Graph Transformer module that embeds final-time propagation probabilities as structural biases in self-attention mechanism, and (ii) a Graph Recurrent Module that captures temporal evolution patterns with bidirectional recurrent networks. Extensive experiments on benchmark graph classification datasets demonstrate that CTQWformer outperforms graph kernel and GNN-based methods, demonstrating the potential of integrating quantum dynamics into trainable deep learning frameworks for graph representation learning. To the best of our knowledge, CTQWformer is the first hybrid CTQW-based Transformer, integrating CTQW-derived structural bias with temporal evolution modeling to advance graph learning.

## 1 INTRODUCTION

Graph-structure data is ubiquitous, and widely used in various domains including social networks, bioinformatics, computer vision. Graph Neural Networks (GNNs) have emerged as a powerful tool for graph learning tasks over graph-structure data, primarily through message passing and neighborhood aggregation mechanisms Wu et al. (2020). Various GNN models have been developed to address different node-level or graph-level tasks. Among these, graph classification constitutes a fundamental graph-level problem that critically depends on accurately modeling the underlying topological structure and the inter-node relationships within graphs.

Inspired by the successful application of Transformers in various domains, researchers have explored using Transformers framework for graph learning, owing to their parallel computation efficiency and strong capability in modeling long-range dependencies. These efforts include directly modeling graph structure or incorporating with GNNs to enhance the performance of various tasks. For instance, Graph Transformer Networks is proposed to learn new graph structures via attention-like modules and then apply conventional GNNs in node classification task Yun et al. (2019). Similarly, Ying et al proposed Graphormer Ying et al. (2021), which apply Transformer architectures to graph data by incorporating different structural encodings into the attention mechanism in graph classification task.

However, traditional GNNs are often limited by their local receptive fields and suffer from issues such as over-smoothing problem and inability to capture long-range dependencies among nodes Corso et al. (2024). Despite the integration of global attention mechanisms and structural biases, Transformer-based GNN models often struggle to effectively capture the intrinsic topological structure and global dependencies inherent in graph data.

In parallel, quantum computing provides a novel computational paradigm via its natural capability for parallel information processing. As the quantum analogue of classical random walks, quantum walks have emerged as a powerful framework for graph learning, owing to their dynamic evolution

on graphs that captures rich and intricate structural information Bai et al. (2015). Specifically, governed by the Schrödinger equation, continuous-time quantum walks (CTQW) offer a powerful and physically grounded way to model graph structure through its dynamical evolution. The constructive and destructive interference effects inherent in CTQW lead to more complex dynamic behaviors in the diffusion process, reflecting local and global properties of graphs, providing insightful structural information beyond its classical features Aharonov et al. (2001).

In this paper, we propose a hybrid graph learning framework, CTQWformer, which integrates CTQW-based physical structural bias and temporal evolution into a unified framework for graph classification task. Our model consists of three core components: (1)The Quantum Walk Encoder (QWE) achieves the dynamical evolution of CTQW over graph structures by modeling node-wise propagation probabilities under different configurations within graph datasets, guided by a trainable Hamiltonian that integrates both underlying graph topology and node features. (2) The Quantum Walk-Graph Transformer (QWGT) module incorporates the final-time propagation probabilities as intrinsic physical structural biases to guide the attention mechanism in graph Transformer. (3) The Quantum Walk-Graph Recurrent (QWGR) module employs a bidirectional recurrent network to processes the dynamic temporal evolution of CTQW, capturing the temporal sequence of node-wise propagation probability among graph. Together, these modules enable CTQWformer to effectively integrate both static physical structural biases and dynamic temporal evolution patterns of CTQW on graphs for graph-level representation learning.

As a result, the contribution of the paper is summarized as follows. First, we design a parameterized Hamiltonian that incorporates both graph structure and node features, which makes the dynamic evolution of CTQW not only rely on the static graph structure, but adaptively capture the connections of node features. The mechanism allows the model to be more suitable for the downstream tasks such as attributed graph classification. Second, we propose the CTQWformer model, a hybrid graph learning framework based on CTQW for graph classification. Specifically, the QWE module implements a trainable CTQW guided by a learnable Hamiltonian that integrates both graph structure and node features, enabling the extraction of dynamical evolution information under various configurations on graphs. The QWGT module encodes static physical structure bias generated by CTQW into attention mechanism in graph Transformer. And the QWGR module utilize bidirectional recurrent neural network to process the dynamic evolution information of CTQW. Together, the proposed module is capable of capturing both static and dynamic evolution information of CTQW on graphs, providing richer information of graph structure and more discriminative representation for graph-level tasks. Third, experiments conducted on several benchmark graph classification datasets demonstrate the effectiveness of our model, achieving competitive performance compared to state-of-the-art methods.

## 2 RELATED WORKS

### 2.1 GRAPH LEARNING

Early approaches to graph learning primarily relied on graph kernel methods, which measure the similarity between graphs by mapping their structure information into a high dimensional Hilbert space Kriege et al. (2020). A wide range of graph kernels have been proposed, many of which fall under the R-convolution framework Haussler et al. (1999), graph similarity is computed by comparing substructures such as walks, subgraphs, and subtrees. Representative examples include the Graphlet kernel Shervashidze et al. (2009) and the Weisfeiler-Lehman subtree kernel Shervashidze et al. (2011). In addition, recent works have explored information-theoretic graph kernels, which measure graph similarity from the perspective of entropy. Several information-theoretic graph kernels have been proposed, such as JTQK Bai et al. (2014), QJSK Bai et al. (2015), HAQJSK Bai et al. (2024), and AERK Cui et al. (2023), all of which are defined based on CTQW and aim to capture both structural and dynamical properties of graphs from the perspective of entropy. These kernels demonstrate the effectiveness of quantum walks as a powerful and efficient tool for graph learning. Although effective in small-scale datasets, kernel methods often suffer from poor scalability and limited expressive power. Typically, a graph kernel is a semi-definite function defined as

$$k\left(G_1, G_2\right) = \langle \phi\left(G_1\right), \phi\left(G_2\right) \rangle \tag{1}$$

Where $G_1$ and $G_2$ are two graphs, $\phi(\cdot)$ denotes the mapping from the input space to a reproducing kernel Hilbert space, and $\langle \cdot, \cdot \rangle$ could be the inner product in the space.

In recent years, GNNs have become the dominant paradigm for graph learning Corso et al. (2024). GNNs learn node and graph representations via neighborhood aggregation mechanism, enabling message-passing across the graph. Typically, GNN perform neighborhood aggregation using spectral-based approaches such as GCN Kipf & Welling (2016), or spatial-based approaches such as GraphSAGE Hamilton et al. (2017) and GIN Xu et al. (2018). Specifically, GCN updates node features by linearly combining the normalized features of their neighbors. GraphSAGE samples a fixed-size set of neighbors and aggregates their features via functions such as mean, LSTM or pooling, enabling inductive learning on large-scale graphs. GIN is designed to achieve maximal expressive power among message-passing GNNs, matching the discriminative capacity of the Weisfeiler-Lehman graph isomorphism test. However, traditional GNNs struggle with long-range dependencies and often suffer from over-smoothing with the increasing depth of networks. Further, to address these limitations, inspired by the success of self-attention mechanism in natural language processing, graph Transformer models have been developed such as Graphormer Ying et al. (2021), GraphGPS Rampášek et al. (2022). However, these models still face challenges: their ability to explicitly model local interactions remains limited, and their interpretability is relatively weak. These drawbacks motivate our approach, which leverages physically grounded quantum walk dynamics to provide both richer local structural modeling and improved interpretability. Formally, the self-attention mechanism in the Transformer framework Vaswani et al. (2017) is defined as

$$\text{Attention}\,(Q,\ K,\ V) = \text{softmax}\left(\frac{QK^T}{\sqrt{d}}\right)V \tag{2}$$

Where $Q,\ K,\ V \in R^{n \times d}$ are the query, key and value matrices derived from the input features respectively. In graph Transformers the input features are node features or edge features, and $d$ is the dimensionality scaling factor.

## 2.2 CONTINUOUS-TIME QUANTUM WALK

Recently, quantum walks, as a general framework for designing quantum algorithms in quantum computing, have demonstrated substantial potential in addressing a variety of graph learning tasks including graph classification Bai et al. (2015), node classification Yan et al. (2022) and link prediction Goldsmith et al. (2023).

Unlike classical random walks, quantum walks leverage superposition and interference to generate fundamentally different propagation behaviors. The dynamical evolution of quantum walks on graphs offers a distinctive perspective for capturing both structural and dynamical properties of graphs, such as probability distribution, spectral of density matrix and von Neumann entropy. Generally, there are two kinds of quantum walks: continuous-time quantum walks and discrete-time quantum walks (DTQW), both of which have demonstrated substantial potential in various graph learning algorithms including graph isomorphism, link prediction and graph kernels Kadian et al. (2021).

Motivated by GQWformer, which pioneers the integration of DTQW and graph Transformers in graph learning Yu et al. (2024). In this paper, we utilize CTQW as it does not require additional coin operators, resulting in more natural and analytically tractable dynamics. This coin-free and analytically tractable formulation not only reduces the effective Hilbert space, simplifies the evolution process but also enables a more expressive and physically grounded encoding of graph structures and node features. By employing a learnable Hamiltonian that fuses graph structure and node features, the trainable CTQW provides a more physically grounded and flexible framework for graph learning.

Specifically, for a graph $G = (V,\ E), n = |V|$ denotes the number of nodes of the graph. In CTQW, the state space of the walker is a Hilbert space $H = span\{|1\rangle,\ |2\rangle,\ \cdots,\ |n\rangle\}$, which is spanned by the position basis states corresponding to the vertices of the graph. The state of the walker is described by a complex vector $|\psi\rangle = [\alpha_1,\ \alpha_2,\ \cdots,\ \alpha_n]$, where $\alpha_i$ denotes the probability amplitude at vertex $i$. As a result, the quantum state $|\psi(t)\rangle$ of CTQW at time $t$ is described as a complex linear combination of these basis states over all vertices.

$$|\psi(t)\rangle = \sum_{v \in V} \alpha_v(t)|v\rangle \tag{3}$$

Figure 1: Illustration of the proposed CTQWformer model. The model integrates CTQW dynamics into graph learning via three core components: (1) QWE simulates trainable CTQW to encode graph structure and features into a time-evolution tensor; (2) QWGT uses final-time propagation probabilities as structural biases in a Transformer; (3) QWGR employs BiGRU to capture node-level temporal dynamics. QWGT and QWGR are fused per layer, enabling deep stacked learning. A global mean pooling and classifier produce the final graph-level prediction.

The equation satisfies $\sum_{v \in V} \alpha_v(t)\alpha_v^*(t) = 1$ for all nodes at any time t. $\alpha_v^*(t)$ is the complex conjugate of $\alpha_v(t)$. The time evolution of the quantum walker is governed by unitary operator $U(t) = e^{-iHt}$. Where $i$ is imaginary unit, $H$ is the Hamiltonian that encodes the graph structure, usually taken from the adjacency matrix $A$ or Laplacian matrix $L = D - A$, where $D$ is the degree matrix, in this paper, we adopt the Laplacian matrix. Finally, the dynamics of the system are described by the Schrödinger equation

$$i\frac{d}{dt}|\psi(t)\rangle = H|\psi(t)\rangle \tag{4}$$

Given initial state $|\psi(0)\rangle$ and time $t$, the state of walker is described as follow

$$|\psi(t)\rangle = U(t)|\psi(0)\rangle = e^{-iHt}|\psi(0)\rangle \tag{5}$$

After the state evolution, we perform a measurement to get the probability distribution of the walker over the graph at time $t$, the probability of the walker at node $i$ is given by

$$p_i(t) = |\langle i|\psi(t)\rangle|^2 \tag{6}$$

Our work builds upon this line of research by integrating CTQW-based features into a trainable graph learning framework, allowing for joint learning of dynamics of CTQW and task-specific graph learning.

## 3 METHOD

### 3.1 THE OVERVIEW OF THE CTQWFORMER MODEL

Formally, given a graph dataset $\mathcal{G} = \{G_1, G_2, \cdots, G_N\}$ where each graph $G_i$ is associated with a graph label $y_i \in R$ and node features $X_i \in R^{n \times d}$, the goal of graph classification is to learn a function that maps an input graph to its corresponding label.

We propose CTQWformer, a novel architecture for graph classification that integrates both static and dynamic information derived from CTQW. The model combines a graph Transformer to capture physically grounded structural bias and a graph recurrent network to model the temporal evolution patterns embedded in the dynamics of CTQW. The proposed model consists of three main components. First, we perform CTQW on graphs to extract encoded information. A learnable Hamiltonian is constructed by integrating graph structure and node features making CTQW trainable and the model to be attribute-aware. Meanwhile, the dynamics evolution of CTQW captures non-local dependencies and global topological features embedded in the graphs. Finally, we perform measurement at discrete time steps $\{t_1, t_2, \cdots, t_T\}$ to get CTQW evolution tensor $P \in R^{T \times n \times n}$ that

encodes the structural and dynamical information of the graphs. The tensor is used in two ways: (1) the final time step probability matrix $P^T$ serves as a structural bias for the graph Transformer; (2) the full tensor $P$ is passed to graph recurrent networks for temporal modeling. Second, we employ a graph Transformer that integrates CTQW-based information to capture structural dependencies. Specifically, extracted from evolution tensor $P$, the final time step probability matrix $P^T$ is designed as a structural bias matrix within the self-attention mechanism to guide the learning of pairwise node interactions. Third, we employ graph recurrent networks based on bidirectional gated recurrent unit (BiGRU) to capture the temporal dynamics of CTQW. The evolution tensor $P$ consists of a temporal sequence of node-pair probability matrices, from which we extract node-wise temporal propagation probabilities from the diagonal elements of node-pair probability matrices, enabling the model to capture dynamic propagation patterns of individual nodes over time.

We integrate the QWGT and QWGR modules into a unified layer, making the architecture suitable for deep graph learning through multi-layer stacking. In each layer, the outputs derived from the QWGT module and the QWGR module are concatenated and passed through a fusion network that effectively combines the static physical structural bias and dynamic temporal evolution features to update node representations. Finally, after multiple stacked layers, a global mean pooling operation is applied on the resulting node embeddings to derive a graph-level representation. The representation is subsequently passed through a final classifier to perform graph-level prediction. The framework of the model is exhibited in Fig.1.

## 3.2 THE QUANTUM WALK ENCODER

The core of our model lies in encoding CTQW-based static physical structural bias and dynamical evolution information into the graph representation. To achieve this, we first construct a trainable Hamiltonian that governs the dynamics of CTQW on the graph. This allows the quantum evolution process to be optimized jointly with downstream tasks, making the model structure-aware and feature-aware.

Specifically, the Hamiltonian $H \in R^{n \times n}$ is designed to integrate both the graph topology and node features. Given a graph $G = (V, E)$ with $n$ nodes and node features $X \in R^{n \times d}$, where $d$ is the dimension of node features. We define a learnable Hamiltonian to construct a trainable CTQW, allowing the model to flexibly adjust to varying graph topologies and node features during the dynamical evolution process. For each edge $(i, j) \in E$, we concatenate the features of two incident nodes to form an edge feature vector

$$e_{ij} = [x_i \| x_j] \in R^{2d} \tag{7}$$

A two-layer perception with non-linear activation is applied to map the edge features into scalar edge weights.

$$w_{ij} = \sigma(W_2 \cdot \phi(W_1 e_{ij})) \tag{8}$$

Where $\phi(\cdot)$ is ReLU nonlinear activation function and $\sigma(\cdot)$ denotes the sigmoid function to ensure the weights are bounded in [0,1]. These learned edge weights form a new weighted adjacency matrix. To ensure the Hermitian property required by the evolution operator, we get symmetrized matrix

$$A_{sym} = \frac{1}{2} \left( A + A^T \right) \tag{9}$$

The final Hamiltonian is defined as a Laplacian matrix

$$H = D - A_{sym} \tag{10}$$

Where $D$ is the degree matrix of $A_{sym}$ with $D_{ii} = \sum_j (A_{sym})_{ij}$. This construction allows CTQW trainable and to be trained jointly with downstream tasks.

Given the learned Hamiltonian, we consider an initial state set consisting of orthonormal basis states, represented by the identity matrix $I_n$, where each column corresponds to an initial state concentrated on a single node, and a set of time steps $\mathcal{T} = \{1, 2, \ldots, T\}$. Subsequently, we can simulate the state evolution of CTQW under various configurations using Schrödinger equation. By stacking the probability distribution under $n$ single-node initial states over $T$ distinct time steps, we obtain the temporal evolution tensor $P \in R^{T \times n \times n}$, which capture the temporal dynamics of the graph. The tensor provides a rich and physically grounded representation of the graph, and thus can be used in downstream models such as graph Transformers and recurrent network for graph-level tasks.

### 3.3 The Quantum Walk-Graph Transformer Module

Although graph Transformers offer a global receptive field that enables comprehensive message passing among all nodes, the self-attention mechanism primarily captures semantic similarity and often neglects the inherent structure properties. To compensate for this, we design The Quantum Walk-Graph Transformer to integrate structural prior information derived from CTQW into the attention mechanism, enabling the model to account for both semantic and structural relations between nodes. Specifically, we utilize the final-time transition probabilities $P^T \in R^{n \times n}$ from CTQW as a static structural bias matrix, which is normalized and log-scaled before being added to the self-attention score matrix.

$$\text{Attention}(Q, K, V, B) = \text{softmax}\left(\frac{QK^T}{\sqrt{d}} + B\right)V \tag{11}$$

Where $Q = XW^Q, K = XW^K, V = XW^V$ are the projected query, key and value matrices. The structural bias $B$ is derived from CTQW probability matrix $P^T \in \mathbb{R}^{n \times n}$ at final time step $T$. To ensure stability, $P^T$ is column-normalized into a stochastic matrix and then transformed as $B = \log(1 + P^T)$.Then node representation is updated and aggregated to produce the CTQW-based graph-level representation $O_{QWGT}$. By integrating the inherent physically grounded structural bias into the attention mechanism, the model gains a richer understanding of graph topology beyond pure node feature similarity.

### 3.4 The Quantum Walk-Graph Recurrent Module

In contrast to the convergence behavior of classical random walks, the evolution of quantum walks is dynamic and oscillatory. These fluctuations encode rich temporal and structural patterns that cannot be captured by static structural bias alone. To model the dynamic evolution of quantum states, we design The Quantum Walk-Graph Recurrent Module, which processes the full CTQW probability sequence over multiple time steps.

We treat the temporal evolution tensor $P \in R^{T \times n \times n}$, generated by evolving each of $n$ single-node initial states over $T$ discrete time steps, as a temporal input tensor. We extract probability matrix for each single-node initial state $|i\rangle$ over $T$ discrete time steps $P_i = \left[P_i^1, P_i^2, \cdots, P_i^T\right] \in R^{n \times n}$, Where $P_i^t$ denotes the probability distribution of CTQW under initial state $|\psi(0)\rangle = |i\rangle$ at time $t$. The sequence is first transformed into a hidden representation via a linear layer and then fed into a BiGRU to model the temporal evolution and fluctuations of each node, enabling the model to capture complex temporal dependencies in both forward and backward directions.

$$H_{GRU} = \text{BiGRU}(\text{Linear}(P_i)) \tag{12}$$

These resulting node representations are aggregated using mean pooling and followed by a feedforward network to generate graph-level representation $O_{QWGR}$.

$$O_{QWGR} = \text{FFN}(\text{MeanPool}(H_{GRU})) \tag{13}$$

This module effectively models the dynamic evolution of CTQW, providing a complementary perspective to the static structural view offered by the QWGT module.

### 3.5 Fusion and Prediction

we construct the CTQWformer layer by integrating the QWGT module and the QWGR module. The QWGT module generates graph-level representations by leveraging CTQW-based structural biases through attention mechanism in the graph Transformer, while the QWGR module extracts graph-level embeddings from the temporal evolution tensor using a BiGRU network. The outputs from these two modules are concatenated and passed through a feed-forward fusion network to produce a unified representation. By stacking multiple CTQWformer layer, the model progressively refines graph representations. At each layer, the graph-level embedding is broadcast back to node-level representations to guide subsequent layers. The final node embeddings from the last layer are aggregated via global mean pooling operation to obtain the final graph-level representation and then fed into a multi-layer classifier for prediction.

| Dataset | MUTAG | PTC(MR) | PROTEINS | DD | IMDB-B | IMDB-M |
|---|---|---|---|---|---|---|
| # Graphs | 188 | 344 | 1113 | 1178 | 1000 | 1500 |
| # Classes | 2 | 2 | 2 | 2 | 2 | 3 |
| Max # Vertices | 28 | 109 | 620 | 5748 | 136 | 89 |
| Mean # Vertices | 17.93 | 14.29 | 39.06 | 284.32 | 19.77 | 13.00 |
| # Node Features | 7 | 18 | 3 | 89 | 0 | 0 |
| Description | Bio | Bio | Bio | Bio | SN | SN |

Table 1: Statistics of benchmark graph datasets.

| Model | MUTAG | PTC(MR) | PROTEINS | DD | IMDB-B | IMDB-M |
|---|---|---|---|---|---|---|
| GCGK | 81.58±2.11 | 57.26±1.41 | 71.67±0.55 | 78.45±0.26 | 65.87±0.98 | 43.89±0.38 |
| WLSK | 82.05±0.36 | 57.97±0.49 | 74.68±0.49 | 79.78±0.36 | 73.40±4.63 | 49.33±4.75 |
| JTQK | 85.50±0.55 | 58.50±0.39 | – | 79.89±0.32 | – | – |
| QJSK | 82.72±0.44 | 56.70±0.49 | 70.13±4.88 | 77.68±0.31 | 62.10 | 43.24 |
| HAQJSK | 85.83±0.72 | 62.35±0.51 | – | – | 73.50±0.45 | **50.08±0.20** |
| AERK | 88.55±0.43 | 59.38±0.36 | – | 77.60±0.47 | – | – |
| **CTQWformer** | **92.54±5.39** | **69.16±5.17** | **78.53±2.34** | **81.24±3.43** | **76.40±1.91** | 47.47±7.84 |

Table 2: Graph classification results (% ± standard deviation) comparing with graph kernel methods. Best scores are in bold.

# 4 EXPERIMENTS

We conduct extensive experiments for graph classification on several benchmark datasets from the TU collection Morris et al. (2020), covering domains including bioinformatics (Bio) and social networks (SN). The detailed statistical information of these datasets is summarized in Table 1. To ensure that all datasets have meaningful node features, we preprocess the graphs by appending normalized node degree using log-scaled max normalization to existing features if available, or using it as the sole node feature otherwise. This approach is commonly adopted in GNN literature You et al. (2020) to enrich structural information in datasets lacking node features such as IMDB-B, IMDB-M Yanardag & Vishwanathan (2015). To evaluate the performance of our proposed CTQWformer, we compare it with two major categories of baseline methods: (1) graph kernel methods and (2) graph neural network approaches.

## 4.1 COMPARISONS WITH GRAPH KERNEL METHODS

**Baseline and Experimental Settings.** We compare the proposed CTQWformer with six graph kernels, including two classical R-convolution graph kernels, (1) the Graphlet Count Graph Kernel (GCGK) Shervashidze et al. (2009) and (2) the Weisfeiler-Lehman Subtree Kernel (WLSK) Shervashidze et al. (2011); four information-theoretic graph kernels, (3) Jensen-Tsallis q-difference Kernel (JTQK) Bai et al. (2014), (4) Quantum Jensen-Shannon Kernel (QJSK) Bai et al. (2015), (5) Hierarchical-Aligned Quantum Jensen-Shannon Kernels (HAQJSK) Bai et al. (2024), (6) Aligned Entropic Reproducing Kernels (AERK) Cui et al. (2023). Notably, all four of these information-theoretic graph kernels are built upon CTQW, demonstrating the potential of CTQW for graph learning. However, the inherent limitations of kernel-based methods prevent them from fully leveraging the rich dynamical evolution information generated by CTQW on graphs.

**Experimental Results and Analysis.** As shown in Table 2, CTQWformer consistently outperforms existing kernel-based methods, including both R-convolution graph kernels and CTQW-based graph kernels, demonstrating its superior ability in capturing graph-level representations. In particular, CTQWformer achieves the best performance on five out of six datasets, except for IMDB-M. This may result from the lack of sufficiently informative node features. While prior CTQW-based kernels already demonstrated the potential of CTQW for graph learning, their performance is limited by the inherent nature of kernel methods. In contrast, CTQWformer integrates the dynamical evolution of CTQW into a trainable representation learning framework, enabling the model to effectively leverage temporal evolution information of CTQW. These results validate the effectiveness of integrating graph neural networks with the dynamical information derived from CTQW on graphs, enabling the model to better capture and exploit the dynamics information of CTQW for graph learning.

| Model | MUTAG | PTC(MR) | PROTEINS | DD | IMDB-B | IMDB-M |
|---|---|---|---|---|---|---|
| GIN-0 | 89.40±5.60 | 64.60±7.00 | 76.20±2.80 | – | 75.10±5.10 | **52.30±2.80** |
| DGCNN | 85.83±1.66 | 58.59±2.47 | 75.54±0.94 | 79.37±0.94 | 70.03±0.86 | 47.83±0.85 |
| PSCN | 88.95±4.37 | 62.29±5.68 | 75.00±2.51 | 76.27±2.64 | 71.00±2.29 | 45.23±2.84 |
| GAT | 89.40±6.10 | 66.70±5.10 | 74.70±2.20 | – | 70.50±2.30 | 47.80±3.10 |
| GCN | 87.20±5.11 | 62.10±1.80 | 75.65±3.24 | 79.12±3.07 | 73.30±5.29 | 51.20±5.13 |
| CAPSGNN | 86.67±6.88 | 66.01±5.91 | 76.40±4.17 | 77.62±4.99 | 71.69±3.40 | 48.50±4.10 |
| GraphSAGE | 79.80±13.9 | 63.90±7.70 | 65.90±2.70 | 65.80±4.90 | 72.40±3.60 | 49.90±5.00 |
| **CTQWformer** | **92.54±5.39** | **69.16±5.17** | **78.53±2.34** | **81.24±3.43** | **76.40±1.91** | 47.47±7.84 |

Table 3: Graph classification results (% ± standard deviation) comparing with GNN-based methods. Best scores are in bold.

## 4.2 COMPARISONS WITH GNN APPROACHES

**Baseline and Experimental Settings.** We compare CTQWformer against seven GNN approaches. The GNN-based baselines consist of (1) Graph Isomorphism Network (GIN-0) Xu et al. (2018), (2) Deep Graph Convolutional Neural (DGCNN) Zhang et al. (2018a), (3) PATCHY-SAN Convolutional Neural Network (PSCN) Niepert et al. (2016), (4) Graph Attention Network (GAT) Veličković et al. (2017), (5) Graph Convolutional Network (GCN) Kipf & Welling (2016), (6) Capsule Graph Neural Network (CAPSGNN) Xinyi & Chen (2018), and (7) Graph Sample and Aggregation (GraphSAGE) Hamilton et al. (2017). These baselines encompass convolutional, attention-based, sorting-based, and capsule-based GNN models, providing a comprehensive comparison with our proposed quantum walk-inspired framework. We follow the standard 10-fold cross-validation setting for all datasets, where accuracy is reported as the mean and standard deviation over 10 folds. Unless otherwise specified, we set the number of CTQWformer layer to 2, and the time steps to 4, the hidden dimension is fixed to 64, and dropout is set to 0.3. We use Adam optimizer with a learning rate of 0.001. And train model for 300 epochs with early stop technique, we slightly adjust the number of layers and time steps to obtain the best performance, within a small grid search range. For baseline methods, we adopt the results reported from their original papers or widely used benchmark studies in published papers Nguyen et al. (2022); Zhang et al. (2018b) for fair comparison.

**Experimental Results and Analysis.** Table 3 shows that CTQWformer achieves the best or highly competitive performance across all datasets except IMDB-M. A potential reason is that the dataset does not provide original node features and consists of relatively small graphs, making it challenging for models like CTQWformer that rely on meaningful node feature information. Nevertheless, the results clearly indicate that CTQWformer successfully integrates CTQW into GNN approaches, providing consistent improvements over conventional GNN baselines. These results demonstrate the effectiveness of incorporating CTQW-based structural priors and temporal information into a trainable deep learning architecture. Unlike traditional GNNs that rely on local message passing, CTQWformer benefits from a global structural perspective via dynamical evolution of CTQW on graphs, which allows the model to better capture complex topological and effectively incorporate node features in graphs.

## 4.3 THE FURTHER ANALYSIS FOR CTQWFORMER

| Model | MUTAG | PTC(MR) | PROTEINS |
|---|---|---|---|
| **CTQWformer** | **92.54±5.39** | **69.16±5.17** | **78.53±2.34** |
| w/o QWGT | 89.39±5.74 | 59.62±4.87 | 77.72±2.41 |
| w/o QWGR | 74.97±7.17 | 57.87±3.25 | 69.00±5.49 |

Table 4: Ablation study of CTQWformer on three datasets.

**Ablation Study.** To assess the contribution of each module in CTQWformer, we perform ablation studies by removing the QWGT module and the QWGR module respectively. The results are reported in Table 4. As shown in the table, we find that removing the QWGR module leads to a significant performance drop across all three test datasets, with accuracy on MUTAG, for instance, decreasing from 92.54% to 74.97%. This highlights the critical importance of modeling dynamical evolution of CTQW to learn graph representations. In contrast, removing the QWGT module results

in a small performance decrease, indicating that while CTQW-based structural bias contributes positively, it plays a less dominant role compared to temporal evolution modeling. Meanwhile, we also observe relatively large standard deviations in some cases, which may be attributed to the inherent dynamics and fluctuation of CTQW evolution, further investigation into these properties remains an important direction for future work.

**Hyperparameter Analysis.** We further investigate the sensitivity of key hyperparameters of CTQWformer, focusing on both the time steps of CTQW and the network depth of the model.

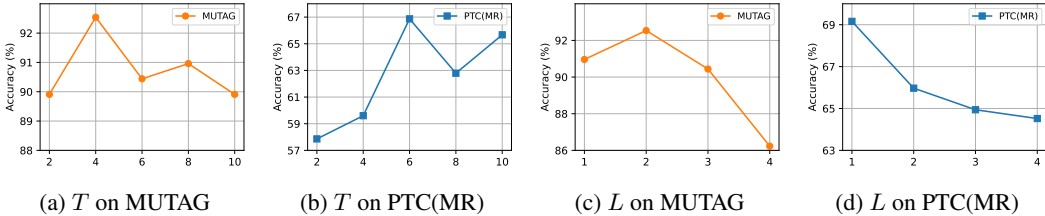

| (a) $T$ on MUTAG | (b) $T$ on PTC(MR) | (c) $L$ on MUTAG | (d) $L$ on PTC(MR) |

Figure 2: Sensitivity analysis of time steps $T$ and network depth $L$.

**Time Steps in CTQW.** Since the dynamics of CTQW is adaptively guided by the learned Hamiltonian, and simulated by traversing all single-node initial states on the graph. We vary the number of time steps $T \in \{2, 4, 6, 8, 10\}$ to study the model's sensitivity to the temporal granularity of CTQW on MUTAG and PTC(MR) datasets. We observe that the classification accuracy on the MUTAG dataset increases with the number of time steps at first and reaches its peak at time steps $T = 4$ (92.54%), indicating that moderate evolution time in CTQW captures informative structural patterns. However, further increasing the time steps leads to a decline in performance, likely due to quantum interference effects or over-smoothing, suggesting that an overly long CTQW evolution may dilute discriminative information. While on the PTC(MR) dataset, increasing time steps from 2 to 6 improves accuracy significantly, indicating better temporal evolution feature capture. Beyond 6 steps, accuracy fluctuates and slightly drops, suggesting excessive steps may introduce redundancy. Thus, 6 time steps offer the best balance between performance and complexity.

**Number of Layers in CTQWformer.** Besides, We vary the number of stacked CTQWformer layers $L \in \{1, 2, 3, 4\}$ to study the effect of network depth. As shown in Figure 3, On MUTAG dataset, the model achieves the highest accuracy with a 2-layer network, outperforming shallower and deeper configurations. This suggests that moderate depth balances representation power and training stability, while excessive depth may cause overfitting or optimization difficulties. While on PTC(MR) dataset, accuracy consistently decreases as the network depth increases from 1 layer to 4 layers, indicating that deeper networks may lead to over-smoothing or overfitting on this smaller datasets, where node representations become indistinguishable and less informative. Overall, these findings highlight the critical importance of judiciously configuring model depth in alignment with the structural complexity and scale of datasets. Empirical evidence from both datasets indicates that a moderate network depth offers a favorable trade-off between expressiveness and generalization, while overly deep architectures tend to diminish performance, particularly in scenarios involving smaller graphs.

## 5 CONCLUSION

In this paper, we have proposed CTQWformer, a novel CTQW-based Transformer model for graph classification task. Our model realizes trainable CTQW on graphs by integrating both graph structure and node features into a learnable Hamiltonian, enabling the model to capture rich and intricate graph structure information. Furthermore, the model incorporates a graph Transformer and a recurrent neural network, which are respectively designed to leverage static physical structural bias and dynamic temporal evolution patterns derived from CTQW. Extensive experiments on multiple benchmark datasets demonstrate the effectiveness and superiority of our method, highlighting the potential of integrating quantum walk dynamics with graph neural networks for graph learning.

## REPRODUCIBILITY STATEMENT

We have made significant efforts to ensure the reproducibility of our results. The full implementation of CTQWformer, along with data preprocessing scripts and instructions for running the experiments, is provided in the supplementary materials.

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

# A COMPLEXITY ANALYZE OF CTQWFORMER

We analyze the computational and memory costs of the core CTQW computations used in CTQW-former and summarize practical trade-offs for scaling.

**Exact matrix-exponential.** Given a graph with $n$ nodes and a Hamiltonian $H \in \mathbb{R}^{n \times n}$, computing the matrix exponential $U(t) = \exp^{-i\mathbf{H}t}$ by direct dense methods (e.g., scaling-and-squaring + Padé) requires $\mathcal{O}(n^3)$ time and $\mathcal{O}(n^2)$ memory per time step. If we compute $U(t)$ at $T$ distinct time points independently, the worst-case time complexity is

$$\text{Time}_{\text{dense}} = \mathcal{O}(T \cdot n^3), \qquad \text{Space}_{\text{dense}} = \mathcal{O}(n^2),$$

and then we store the full evolution tensor $P \in \mathbb{R}^{T \times n \times n}$ the space grows to $\mathcal{O}(T \cdot n^2)$.

**Memory trade-offs and what to store.** CTQWformer uses CTQW-derived information in two ways: (i) the final-time structural bias $P_T$ for the Graph Transformer module, and (ii) temporal sequences for the Graph recurrent module. Storing the full pairwise evolution tensor $P \in \mathbb{R}^{T \times n \times n}$ is often the dominant memory cost ($\mathcal{O}(Tn^2)$).

**Structural Bias of CTQW in The QWGT module** Specifically, given the raw transition probability matrix $P^T$ with entries $p_{ij}^T$, we normalize along the column dimension to obtain a stochastic matrix

$$\tilde{P}_{ij}^T = \frac{p_{ij}^T}{\sum_{i'} p_{i'j}^T}, \tag{14}$$

ensuring that $\sum_i \tilde{P}_{ij}^T = 1$ for each $j$. The structural bias is then defined as

$$B = \log\left(1 + \tilde{P}^T\right), \tag{15}$$

which stabilizes training and smooths the influence of CTQW probabilities when added to the attention logits.

**Temporal Sequence of CTQW in the QWGR module** In addition to the final-time distribution, we also leverage the temporal evolution information of CTQW. The temporal evulution tensor $P$ is treated as a sequence input, where $P^t \in \mathbb{R}^{N \times N}$.

For each node $i$, we extract its temporal sequence from the diagonal entries of $P^t$:

$$s_i = \{p_{ii}^t\}_{t=1}^T, \tag{16}$$

where $p_{ii}^t$ denotes the probability that the walker remains at node $i$ at time step $t$. These node-level time series are projected into a latent space and then fed into a bidirectional GRU (BiGRU) encoder:

$$h_i = \text{BiGRU}(s_i), \tag{17}$$

where $h_i$ is the middle representation of hidden layer capturing both forward and backward temporal dependencies of node $i$. The resulting node embeddings $\{h_i\}_{i=1}^N$ are aggregated by mean pooling:

$$h_G = \frac{1}{N} \sum_{i=1}^N h_i, \tag{18}$$

and further transformed by a feed-forward readout network to obtain the final graph-level embedding.

**Fusion Strategy.** Finally, the two branches operate in parallel and produce graph-level vectors: $O_{\text{QWGT}}$ from the QWGT module and $O_{\text{QWGT}}$ from the QWGR module. They are fused by concatenation:

$$O_{\text{fused}} = [O_{\text{QWGT}} \| O_{\text{QWGR}}],$$

The fused representation is then passed through a feed-forward projection to form the final graph embedding used for classification.

## B  NUMERICAL SIMULATION OF CONTINUOUS-TIME QUANTUM WALK

For the numerical simulations of CTQWs, we utilize PyTorch for the matrix exponential computations, running on an RTX 4090 GPU, with a CPU configuration of 16-core Intel(R) Xeon(R) Gold 6430 and 120GB RAM. Additional implementation and training details can be found in the supplementary materials.

It is worth noting that under the scale of the benchmark datasets used in this work, directly computing the evolution operator $U(t) = e^{-iHt}$ is feasible with current hardware, and the numerical stability is well maintained. More importantly, the design of CTQWformer is not restricted to small graphs. In our implementation, the matrix exponential $e^{-iHt}$ is computed using PyTorch's built-in $torch.matrix\_exp$ , which internally relies on a scaling-and-squaring with Padé approximation method. This ensures differentiability and compatibility with automatic back-propagation. For small to medium-scale benchmarks, we adopt the exact matrix exponential to minimize numerical approximation errors and provide a clean evaluation of the model itself. For larger graphs, these approximations can be seamlessly integrated without altering the framework. In fact, the model only requires applying $U(t)$ to vectors (i.e., computing $U(t)v$), which can be efficiently approximated on large graphs using Krylov subspace or Lanczos methods without explicitly forming the full matrix exponential. Such approximations scale similarly to sparse matrix–vector multiplications, making them applicable to graphs with tens of thousands of nodes or more. Therefore, while our experiments focus on small- to medium-scale graphs, CTQWformer is inherently scalable and can be naturally extended to large-scale graph learning tasks.

## C  ON THE COMPARISON WITH GRAPH TRANSFORMERS

We note that graph Transformer architectures such as Graphormer Ying et al. (2021) and GraphGPS Rampášek et al. (2022). Graphormer enhances Transformer for graphs via integrating structural biases and centrality encoding, while GraphGPS unifies local, global and relative attention to capture multi-scale graph structure information. They are primarily designed for large-scale datasets (e.g., OGB benchmarks), and their scalability and structural encodings show advantages. However, reproducing their results on TU collection datasets is non-trivial, as their original implementations are not tailored for small-scale graph classification tasks and often require heavy hyperparameter tuning to yield stable results. More importantly, our CTQWformer is not a pure graph Transformer: it is a hybrid framework that integrates quantum walk dynamics with graph Transformers and recurrent network, thereby providing complementary modeling of temporal evolution beyond the structural biases used in Graphormer/GraphGPS. We therefore regard these models as orthogonal to our contribution, and leave extensive large-scale comparisons to future work.

## D  THE USAGE DISCLOSURE OF LARGE LANGUAGE MODELS

We used large language models (LLMs), specifically ChatGPT and Deepseek, as assistive tools during the preparation of this paper. The LLM was employed only for language refinement and improving readability, such as rephrasing sentences, polishing grammar, and adjusting writing style, and for code debugging support in the experiments. All research ideas, model design, theoretical analysis, and experimental implementation were conceived and carried out solely by the authors. The LLM did not contribute to the development of research methodology, experimental setup, or interpretation of results. The authors take full responsibility for the content of this paper.

