# OpenReview forum: "CTQWFORMER: A CTQW-BASED TRANSFORMER FOR GRAPH CLASSIFICATION"
_ICLR.cc/2026/Conference — ICLR 2026 Conference Withdrawn Submission_

### Official Review · Reviewer_guxs · 2025-10-17

**Soundness:** 2
**Presentation:** 2
**Contribution:** 3
**Rating:** 2
**Confidence:** 4

**Summary:**

This works introduces an original quantum walk-(QW) based structural encoding for graph transformers (GT).
QW probabilities are computed using a time-dependent evolution operator, mimicking Schrodinger's equation, and later used in self-attention and recurrent modules, providing rich structural information to the network.

Although the idea of using QW as a structural encoding in GT is novel and very interesting, a lack of reflection on the recent GT literature, which comprises closely related works, prevents a fair assessment of the novelty of this work.
Furthermore, the experimental comparison lacks all modern GNN methods and omits GT altogether, which is a serious obstacle for an ICLR publication.

**Strengths:**

The introduction of QW to enhance the expressivity of GT is an exciting research idea, and constitutes an interesting alternative to current structural encoding methods.
QW have the potential to finely encode the structural information carried by graph topology, and their application for graph classification is definitely worth investigating.

**Weaknesses:**

Despite its undeniable originality and potential, the implementation of the main contribution of this work, i.e. the use of QW as a GT structural encoding method, remains very classical and the lack of perspective on very close research raises questions on its actual benefits.

This work is therefore crucially missing a comparison with "classical" random walks (RW) and their use in graph transformers (especially [6] and [7] below), that have already shown state-of-the-art performances on various graph benchmarks. Just like QW, RW richly encodes graph structures at the edge level up to a $T$-hop neighborhood where the maximum walk length $T$ can be adjusted.
In the absence of any mention of these works, the reader cannot measure the benefits of QW over RW in terms of expressivity, scalability or interpretability. This is problematic as this work essentially proposes a replacement for a highly efficient and widely adopted method.

Regarding the lack of originality in the implementation, biasing the self-attention matrix of GT with edge-wise structural information is a classical method used in Graphormer, which the authors cite, but also in all the influential papers referenced below ([1-8]) which the authors do not mention. The "dynamical" aspect of walks, which is encoded here using bidirectional-GRU, is also tackled in an arguably more efficient manner in both GRIT [6] and CSA [7]. In summary, this work presents few novelties beyond its initial idea, and the lack of discussion about close methods does not allow to accurately evaluate its benefits, which are certainly real.

A second, very serious weakness, is the incomplete experimental evaluation. The authors invoke difficulties in re-implementing GT methods, which is understandable, but several of the works listed below provide their code and weights (and so does GraphGPS which is mentionned in the paper), and comparing with GT methods is mandatory when proposing a new GT architecture. I suggest that the authors implement the GT methods that can be the most conveniently trained on their datasets (including GRIT and GraphGPS) to strenghthen their experimental results.

The statement made by the authors that GT are orthogonal to their work is factually wrong, given how close many of these works are to the presented model. It is also contradicted by a statement made in section 2.1, which presents the limitations of current GT as a motivation for this work ("These drawbacks motivate our approach", l122).

The comparison with GNN methods is also piecemeal, and lacks all works past 2018 (such as [9]).

These serious limitations prevent acceptance in the present state of this work.


[1] A generalization of transformer networks to graphs, Dwivedi & Bresson, 20

[2] Rethinking graph transformers with spectral attention, Kreuzer et al., NeurIPS 21

[3] Graphit: Encoding graph structure in transformers, Mialon et al., 21

[4] Global self-attention as a replacement for graph convolution, Hussain et al., KDD 22

[5] Graph propagation transformer for graph representation Learning, Chen et al., IJCAI 23

[6] Graph inductive biases in transformers without message passing, Ma et al., ICML 23

[7] Self-attention in colors: Another take on encoding graph structure in transformers, Menegaux et al., TMLR 23

[8] Enhancing Graph Transformers with Hierarchical Distance Structural Encoding, Luo et al., NeurIPS 24

[9] From stars to subgraphs: Uplifting any gnn with local structure awareness, Zhao et al., ICLR 22

**Questions:**

The clarification of these additional points would be of further help to the reader:

- Section 3.4: "In contrast to the convergence behavior of classical random walks, the evolution of quantum walks is dynamic and oscillatory." This sentence needs additional clarification. The $e^{-t}$ in Equation (5) shows that states reached by QW are on the contrary stationary.

- Equation 13: This part would need further clarification, especially regarding how $O_{QWGR}$ is used in following layers. Encoding the output of this module into a single graph-level representation likely causes severe information loss. Expliciting the dimensions would help the reader comprehend better these operations.
It is not clear how $P_i$ is extracted from $P$ without refering to the appendices. Please check the dimensions of $P_i$.

Several typos / writing issues can be found in the manuscript. This includes:
- Section 2.2, l141: "spectral of density matrix"
- l146: "Motivated by GQWformer, which pioneers the integration of DTQW and graph Transformers in graph learning."
- l250: "a two-layer perception"

---

### Official Review · Reviewer_Y43p · 2025-10-18

**Soundness:** 2
**Presentation:** 2
**Contribution:** 2
**Rating:** 2
**Confidence:** 3

**Summary:**

The paper introduces a new type of graph classification method called CTQWformer that leverages techniques from quantum walks to enhance the performance of attention-based graph representation construction. Experimental results demonstrate competitive performance of CTQWformer over small-to-medium scale graph classification benchmarks.

**Strengths:**

Utilizing ideas from quantum mechanics is an interesting and promising direction toward better design of graph representation methods------as the authors have also stated in the paper, sometimes physically-oriented design might brought better inductive biases that potentially improves tasks where such types of features are not trivially captured through mathematical initiatives like kernel methods or message passing neural networks.

**Weaknesses:**

- **On physically structural bias**: Despite the authors use methods from quantum mechanics, it does not appear to me what the clear signal of strength is for such methods under the context of graph representation learning: Does it enhances expressivity? Is there any problems which cannot be solved by message passing or kernel methods (which, up to Weisfeiler-Lehman hierarchies are somewhat closely related) that are solvable via quantum walks? I think in addition to technical introduction, the authors shall discuss more on this design choice.
- **Lack of empirical comparisons**: While the proposed architecture has close relationship with graph transformers, the authors only compared with kernel methods and message passing methods before 2020 in their empirical investigations. I do not think this is a proper way of doing comparisons in 2025 as graph representation learning is a very active field of research. I also noticed that in appendix C the authors mentioned that due to implementation reasons comparisons with graph transformers are not conducted. However, it appears to me that contemporary graph transformer designs are not that difficult to implement, imo they seem to be simpler than quantum methods.
- **Minor presentation issues**: In equation (9) in line 256, should the presentation be like $W_\text{sym}$ instead of $A_\text{sym}$? The notation is somewhat confusing for me.

**Questions:**

Aside from weaknesses, I have one additional question:
- What is the complexity comparison between CTQWformer and conventional graph transformers? While their is a complexity breakdown in appendix A, which seems not very satisfactory for large graphs. I would like to see a detailed comparison between per-graph computational complexities between CTQWformer and state-of-the-art graph transformer models like GraphGPS.

---

### Official Review · Reviewer_r2V5 · 2025-10-18

**Soundness:** 2
**Presentation:** 3
**Contribution:** 2
**Rating:** 2
**Confidence:** 4

**Summary:**

This paper proposes CTQWformer, a novel hybrid framework for graph classification. It integrates continuous-time quantum walks with a graph transformer and a graph recurrent module. Using a trainable Hamiltonian that fuses topology and node features, the model uses continuous-time quantum walks-derived probabilities as a static structural bias for the transformer's attention while the recurrent module captures the dynamics of the temporal evolution.

**Strengths:**

The paper introduces CTQWformer, a hybrid framework to integrate continuous-time quantum walks (CTQW) with both graph transformer and recurrent modules. It uniquely uses a trainable Hamiltonian to model both static structural bias and dynamic temporal evolution from the CTQW dynamics. Furthermore, the paper is well-organized, and the methodology is presented clearly, making the complex concepts easy to follow.

**Weaknesses:**

1. The model's scalability is severely limited, and the complexity analysis is critically flawed on two fronts. First, the classical computation of the matrix exponential requires $\mathcal{O}(T \cdot n^3)$ time and $\mathcal{O}(T \cdot n^2)$ memory, which is computationally infeasible for large graphs. While the authors suggest using Krylov methods for scaling, this contradicts the model's design, as the QWGT module explicitly requires the full $n \times n$ probability matrix $P^T$ as a structural bias, not just a more efficient matrix-vector product.

2. More importantly, the entire analysis ignores the cost of quantum measurement. To obtain the probability matrix $P$ from the quantum walk, one must perform quantum state tomography, the complexity of which scales exponentially with the number of nodes (qubits). This exponential bottleneck is a well-known barrier in quantum computing, and its omission suggests the proposed "quantum" framework might not practically scalable.

3. A significant weakness is the selection of baselines, which are largely outdated, with most models dating from 2016-2018 (e.g., GCN, GAT, GIN). While it is understood that quantum-based methods are challenging to apply to large-scale datasets, the paper's validation is further weakened by its exclusive focus on graph classification. Reviewer would be happy if the authors can conduct numerical experiments on other general tasks, such as node classification, link prediction or graph reconstruction. Alternatively, performing property prediction on specific small-graph domains on which the quantum-based method can apply, like small molecules, would have provided a more convincing validation of the framework's capabilities beyond a single task.

4. The experimental results demonstrate significant instability, which undermines the reliability of the model's reported performance. The standard deviations are exceptionally high across multiple datasets, such as $92.54 \pm 5.39$ on MUTAG, $69.16 \pm 5.17$ on PTC, and a very large $47.47 \pm 7.84$ on IMDB-M. The authors acknowledge this volatility, attributing it to the "inherent dynamics and fluctuation of CTQW evolution". Such high variance suggests the model is extremely sensitive to random initialization or data splits and is not robust, making the reported mean accuracy a potentially misleading metric of its true performance.

**Questions:**

Please see the weaknesses.

---

### Official Review · Reviewer_ogv6 · 2025-11-01

**Soundness:** 3
**Presentation:** 2
**Contribution:** 3
**Rating:** 6
**Confidence:** 3

**Summary:**

The paper proposes a novel hybrid framework that integrates continuous-time quantum walks (CTQW) with graph neural networks (GNNs) and Transformer architectures to enhance graph classification tasks. The model leverages a trainable Hamiltonian from CTQW to extract and encode both static and structural biases and dynamic temporal evolution patterns. These are incorporated into two modules: a Graph Transformer (QWGT) and a bidirectional recurrent network (QWGR), which are fused to produce rich graph-level representations. Extensive experiments on benchmark datasets demonstrate that CTQWformer outperforms both kernel-based and GNN-based methods.

**Strengths:**

The idea of integrating quantum walk dynamics into a trainable graph transformer is interesting and novel.

The use of a learnable Hamiltonian allows the model to adaptively capture both structural and feature-based relationships in graphs.

The empirical results are compelling, showing consistent improvements across multiple datasets. The ablation studies and sensitivity analyses further validate the contributions of each component and provide insights into optimal configurations.

**Weaknesses:**

While the idea is interesting, the motivation is difficult to grasp. In the Introduction, the authors state that Transformers can enhance GNNs due to their "strong capability in modelling long-range dependencies." However, they later claim that "Transformer-based GNNs often struggle to capture both local and global dependencies in graph data." This appears inconsistent and raises questions about the clarity of the motivation.

The proposed model is tightly coupled with Continuous-Time Quantum Walks (CTQW), and its description heavily relies on CTQW concepts, which makes the model difficult to understand without prior knowledge of CTQW.

The results in Table 2 show high standard deviations, whereas those in Table 3 do not. Is there a specific reason for this discrepancy?

**Questions:**

Please see above comments on weaknesses.

---

### Note · Authors · 2025-11-12

I have read and agree with the venue's withdrawal policy on behalf of myself and my co-authors.